# Analysis of Structure and Antioxidant Activity of Polysaccharides from *Aralia continentalis*

**DOI:** 10.3390/ph15121545

**Published:** 2022-12-13

**Authors:** Yan-bo Hu, Hui-li Hong, Li-yang Liu, Jia-ning Zhou, Yue Wang, Yi-ming Li, Li-yuan Zhai, Zeng-hui Shi, Jun Zhao, Duo Liu

**Affiliations:** 1School of Food Sciences and Engineering, Changchun University, Changchun 130024, China; 2School of Life Sciences, Changchun Normal University, Changchun 130032, China

**Keywords:** *Aralia continentalis*, polysaccharides, structural characterization, antioxidant activities

## Abstract

We extracted, purified, and characterized three neutral and three acidic polysaccharides from the roots, stems, and leaves of *Aralia continentalis* Kitigawa. The results of the analysis of monosaccharide composition indicated that the polysaccharides from the roots and stems were more similar to each other than they were to the polysaccharides from the leaves. The in vitro antioxidant results demonstrated that the acidic polysaccharides had stronger antioxidant activity than the neutral fractions. Therefore, we investigated the primary purified acidic polysaccharide fractions (WACP(R)-A-c, WACP(S)-A-c, and WACP(L)-A-d) by NMR and enzymatic analysis. The structural analytical results indicated that WACP(R)-A-c contained homogalacturonan (HG); WACP(S)-A-c contained HG and rhamnogalacturonan II (RG-II), and WACP(L)-A-d contained HG, RG-II, and rhamnogalacturonan I (RG-I) domains. Our findings offer insights into the screening of natural polysaccharide-based antioxidants and provide a theoretical basis for the application of *A. continentalis.*

## 1. Introduction

*Aralia continentalis* Kitagawa is widely distributed in mainland Asian countries, such as Japan, Korea, and China [1,2]. Masao Kitagawa first identified *A. continentalis* in 1935, and it was originally confined to Japan as a vegetable [3]. Gradually, other Asian countries, such as China and Korea, acknowledged its edible and medicinal values [4]. The dried roots of *A. continentalis* have been widely used in traditional medicine owing to their excellent biological characteristics, such as their anti-inflammatory, antiarthritic, and antibacterial effects [3,5,6]. To date, research has focused on the alcohol extracts obtained from the dried roots of *A. continentalis.* Several reports have suggested that the alcohol extract of the *A. continentalis* root has anti-inflammatory [7,8,9] and anti-nociceptive [10] pharmacological properties. For example, the ethanolic extract of *A. continentalis* roots inhibited cartilage degradation by downregulating MMP activity and chondrocyte apoptosis. Additionally, a diterpene component isolated from the methanol extract of *A. continentalis* exhibited analgesic and anti-inflammatory activity.

Recent studies have demonstrated that plant polysaccharides play a vital role in plant growth and development and have a wide range of antioxidant activities [11,12,13,14]. Polysaccharides, the major active components of *A. continentalis*, are mostly present in water extracts. Increasingly, studies have reported the use of water and enzymatic extraction to obtain the soluble polysaccharides of the *A. continentalis* root, which exhibit a range of bioactivities, including antioxidant [15], antiaging [16], and antibacterial characteristics. Several studies have also reported on the leaves and stems of *A. continentalis* (edible wild herbs). The structural characterization and antioxidant analysis of the polysaccharides from different *A. continentalis* parts could support the development of plant-derived polysaccharides as natural antioxidants. Additionally, this study provides potential strategies for the deep processing of edible wild herb resources.

We extracted polysaccharides from the roots, stems, and leaves of *A. continentalis*; compared their antioxidant activity, sugar composition, and molecular weight; and characterized the structural features of the most potent polysaccharides that displayed antioxidant activity. These results elucidate the structural characteristics of *A. continentalis* polysaccharides and provide a basis for their development for use in functional foods and medicine.

## 2. Results

### 2.1. Extraction of the Polysaccharides from Three Parts of A. continentalis

We extracted three water-soluble polysaccharide-enriched fractions (WACP[R], WACP[S], and WACP[L]) from the roots, stems, and leaves of *A. continentalis* with final yields of 3.6%, 12.9%, and 8.2%, respectively. The yields of WACP(S) were higher than those of WACP(R) and WACP(L). Next, we investigated the chemical characteristics and monosaccharide composition of the polysaccharides. The chemical characterization indicated that the total sugar content of WACP(R) was higher than that of WACP(S) and WACP(L), while the uronic acid content was highest in WACP(S). All three polysaccharides contained low amounts of protein and ash. Polysaccharides from plant cell walls contained pectin and cellulose and were primarily composed of galacturonic acid (GalA), glucose (Glc), galactose (Gal), and arabinose (Ara). The analysis of monosaccharide composition showed that WACP(R) and WACP(S) were primarily composed of Glc, Gal, GalA, and Ara (Table 1), while WACP(L) was composed of Glc, Gal, and GalA. These results indicated that WACP(R) and WACP(S) were more similar to each other in terms of their monosaccharide composition and content than they were to WACP(L).

### 2.2. Fractionation of Polysaccharides from A. continentalis

We fractionated WACP(R), WACP(S), and WACP(L) using anion-exchange chromatography and eluted the neutral fractions of the three polysaccharides (WACP(R)-N, WACP(S)-N, and WACP(L)-N) with distilled water and the acidic fractions (WACP(R)-A, WACP(S)-A, and WACP(L)-A) with 0.5 M NaCl. We determined the monosaccharide compositions of these polysaccharide fractions by high-performance liquid chromatography (HPLC) (Table 2). Both WACP(R)-N and WACP(S)-N were composed of Glc, Gal, and Ara, while WACP(L)-N was composed of Glc (67.4%), with a lower content of Gal (13.1%) and Ara (7.2%). Similarly, WACP(R)-A and WACP(S)-A contained GalA as the major monosaccharide, followed by Gal and Ara, while WACP(L)-A contained GalA (36.1%), Gal (26.7%), and Ara (15.7%) (Table 2). Thus, the neutral polysaccharide from the leaves and the acidic polysaccharides from the roots and stems of *A. continentalis* could be composed of homoglucan and homogalacturonan, respectively.

### 2.3. FTIR Spectra Analysis

We analyzed the neutral and acidic polysaccharide fractions based on their FTIR spectra (Figure 1), which were all similar. The intense absorption band near 3407 cm^−1^ corresponded to the stretching vibration of O-H. The weak band around 2940 cm^−1^ corresponded to the C-H stretching vibration [17]. The peaks around 1747 cm^−1^ and 1613 cm^−1^ represented the C=O stretching vibrations of methyl-esterified and free COO-, respectively [18]. The absorption around 1747, 1613, and 1413 cm^−1^ indicated the presence of uronic acid [19]. As depicted in Figure 1, we observed absorption around 1747 cm^−1^ for the acidic polysaccharide fractions but not for the neutral polysaccharide fractions. Additionally, the characteristic peaks around 826 cm^−1^ suggested the presence of α-linked glycosyl residues [20]. The stretching bands around 1025 to 1147 cm^−1^ indicated C-O bonds [21]. The FTIR results indicated that we had successfully isolated the neutral and acidic polysaccharides and that the acidic polysaccharides were partially esterified.

### 2.4. In Vitro Antioxidant Activity Assays of the Polysaccharides from A. continentalis

We evaluated the antioxidant activity of the six polysaccharides from *A. continentalis* (WACP(R)-N, WACP(S)-N, WACP(L)-N, WACP(R)-A, WACP(S)-A, and WACP(L)-A) based on their ability to scavenge hydroxyl, superoxide anion, and DPPH radicals. As depicted in Figure 2, all three acidic polysaccharides (WACP(R)-A, WACP(S)-A, and WACP(L)-A) displayed dose-dependent antioxidant activity in the range of 0.5 mg/mL to 10 mg/mL in the three assays. However, the three neutral polysaccharides (WACP(R)-N, WACP(S)-N, and WACP(L)-N) had lower activity toward antioxidants except in regard to their ability to scavenge superoxide radicals. Of these acidic polysaccharides, WACP(R)-A displayed the highest antioxidant activity in the DPPH radical-scavenging assay. The DPPH radical-scavenging ratio of WACP(R)-A reached 95.1% at a concentration of 10 mg/mL. Similarly, WACP(S)-A displayed the highest antioxidant activity in the hydroxyl radical-scavenging assay with a clearance rate of 70.8%. Thus, the acidic polysaccharides could be the primary antioxidant functional components extracted from *A. continentalis* and represent potential natural antioxidants.

### 2.5. Structural Analysis of Acidic Polysaccharides from A. continentalis

#### 2.5.1. Purification of Acidic Polysaccharide Fractions

We purified the acidic polysaccharides (WACP(R)-A, WACP(S)-A, and WACP(L)-A) from the roots, stems, and leaves of *A. continentalis* and analyzed their structures owing to their significant antioxidant activity. We separated them further by gel-permeation chromatography (Figure 3A–C). The molecular weights of WACP(R)-A-c, WACP(S)-A-c, and WACP(L)-A-d were 27.3, 9.5, and 15.8 kDa, respectively (Figure 3D–F). We determined the monosaccharide compositions of these polysaccharide fractions by HPLC (Table 3). Both WACP(R)-A-c and WACP(S)-A-c contained GalA as the major sugar, with contents of 84.5% and 83.0%, respectively. WACP(L)-A-d was composed of GalA (44.8%), Gal (19.3%), Ara (12.3%), and Rha (11.5%). We conducted a detailed structural characterization of WACP(R)-A-c, WACP(S)-A-c, and WACP(L)-A-d using enzymolysis and ^13^C NMR spectral analyses.

#### 2.5.2. Enzymological Analysis

Based on the monosaccharide compositions of WACP(R)-A-c and WACP(S)-A-c, we used endo-polygalacturonase (PG) M2, an efficient and specific pectic enzyme for hydrolyzing homogalacturonan (HG) to hydrolyze the polysaccharide fractions. Correspondingly, we used endo-PG, β-1,4-galactanase, and α-L-arabinase to hydrolyze WACP(L)-A-d. As shown in Figure 4, the molecular weights of WACP(R)-A-c, WACP(S)-A-c, and WACP(L)-A-d changed significantly upon the addition of endo-PG to the reaction system. However, the molecular weight distributions of the three purified polysaccharide fractions differed, indicating that their hydrolysates varied. The hydrolysates of WACP(R)-A-c were primarily composed of oligosaccharides and polysaccharide fragments of 5.7 kDa. The hydrolysates of WACP(S)-A-c comprised oligosaccharides and polysaccharide fragments of 5.7, 2.7, and 2.0 kDa. Finally, the hydrolysates of WACP(L)-A-d were more complex than those of WACP(R)-A-c and WACP(S)-A-c, since they included oligosaccharides and polysaccharide fragments of 13.1, 6.3, 2.7, and 2.0 kDa. The link and domain types of the three polysaccharides differed. WACP(R)-A-c and WACP(S)-A-c contained homogalacturonan (HG), while WACP(S)-A-c contained other types of pectin. Compared with that of WACP(R)-A-c and WACP(S)-A-c, the composition of WACP(L)-A-d was more complex, possibly also containing RG-I or RG-II besides HG. Therefore, based on the monosaccharide composition of WACP(L)-A-d, efficient and specific enzymes for hydrolyzing the side chains of RG-I, such as endo-β-1,4-galactanase, and endo-α-1,5-arabinanase, were used to hydrolyze this polysaccharide fraction. As shown in Figure 4C, the molecular weight of WACP(L)-A-d decreased, indicating that the pectin RG-I was present in WACP(L)-A-d.

##### 2.5.3. ^13^C NMR Spectra Analysis

We determined the structural characteristics of the acidic polysaccharide fractions according to their ^13^C NMR spectra. As shown in Figure 5A,B, WACP(R)-A-c and WACP(S)-A-c exhibited similar ^13^C NMR spectra. The signals at 174.99 ppm corresponded to the C-6 of unesterified 1,4-α-D-Gal*p*A, and the signals at 100.34, 67.89, 70.48, 78.96, and 71.32 ppm were attributed to the C-1, C-2, C-3, C-4, and C-5 of 1,4-α-D-Gal*p*A [22]. These signals indicated that HG-type pectin was the major component of these fractions. We also detected the C-6 of methyl-esterified GalA at 170.73 ppm and the methyl group at 52.81 ppm in these fractions [23]. These results suggested that the HG domains in these fractions were methyl esterified. In the WACP(S)-A-c spectrum, except for the existence of methyl-esterified HG-type pectin, the signals at 96.10 and 92.14 ppm corresponded to the anomeric C-2 of α-D-Kdo*p* and the C-2 of α-L-Ace*f*A, respectively [24], which were characteristic of the RG-II domain. We also observed signals for the C-1 and C-6 of 1,4-α-D-Gal*p*A in the WACP(L)-A-d spectrum (Figure 5C). Furthermore, we attributed the signals at 20.49 ppm to the acetyl groups attached to the α-D-Gal*p*A units, confirming the esterification of HG-type pectin in WACP(L)-A-d. We observed the signals of the C-6 of 1,2-α-L-Rha*p* at 16.54 ppm and assigned the signals at 62.45 ppm to the C-6 of the β-D-Gal*p* units [25]. These signals indicated the existence of the RG-I domain. Moreover, we assigned the signals at 96.10 and 92.14 ppm to the anomeric C-2 of α-D-Kdo*p* and the C-2 of α-L-Ace*f*A, respectively, which were characteristic of the RG-II domain. Thus, the ^13^C NMR spectra indicated that WACP(R)-A-c was composed of methyl-esterified HG-type pectin, WACP(S)-A-c was composed of methyl-esterified HG and RG-II domains, and WACP(L)-A-d was composed of methyl/ethyl-esterified HG, RG-I, and RG-II domains. These results were consistent with those of enzymatic hydrolysis.

## 3. Discussion

The three major categories of pectin, based on its structural characteristics, are HG, RG-I, and RG-II [26]. According to our structural analysis, the three polysaccharide fractions from *A. continentalis* contained HG domains. In terms of their monosaccharide compositions, the mass ratios of HG in WACP(R)-A-c and WACP(S)-A-c were similar, and WACP(S)-A-c contained a small amount of RG-II in addition to the HG domains. However, the mass ratio of HG in WACP(L)-A-d was significantly different. In addition, a small amount of HG indicated that WACP(L)-A-d also contained rhamnogalacturonan-I (RG-I) and rhamnogalacturonan-II (RG-II). Thus, the polysaccharides from different parts of *A. continentalis* varied, suggesting possible differences in biological activity. Therefore, we explored the basic biological activity of the polysaccharides from different parts of *A. continentalis*. Polysaccharides exhibit potential antioxidant activity [27,28], particularly when they contain HG-type pectin. HG-type pectin demonstrates stronger antioxidant activity in the presence of carboxyl groups [29]. For instance, the HG-rich pectic polysaccharides from *Lonicera japonica* Thunb and *Radix sophorae Tonkinensis* have higher GalA contents with significant antioxidant activity compared with those of the other polysaccharide fractions [30,31]. Our in vitro antioxidant results showed that the acidic polysaccharides of *A. continentalis* had stronger antioxidant activities than the neutral fractions. Furthermore, the higher content of GalA residues and HG domains in WACP(R)-A-c and WACP(S)-A-c could explain their higher amounts of antioxidant activity compared with WACP(L)-A-d.

We performed enzymological and ^13^C NMR spectra analyses to investigate the structural characteristics of the polysaccharide fractions from the roots, stems, and leaves of *A. continentalis*. The pectin structure in different parts of the same plant and at different growth periods can vary. Our results demonstrated that the roots and stems contained similar pectin types, such as HG-type pectin. The root of *A. continentalis*, used as a traditional Chinese medicine [32], possesses anti-inflammatory [33], antirheumatic [34], and hepatoprotective effects [35]. However, application of the root is limited owing to its low yield and high cost. Therefore, finding an equally effective alternative to *A. continentalis* root is a necessity. Compared with the roots, the stem of *A. continentalis* is a highly abundant and renewable resource. It is also edible, so could be used in functional food and medicine instead of the roots of *A. continentalis*.

## 4. Materials and Methods

### 4.1. Materials

We obtained *A. continentalis* from the School of Landscape Architecture, Changchun University, Changchun, China. We purchased DEAE-cellulose from the Shanghai Chemical Reagent Research Institute (Shanghai, China); Sepharose CL-6B from GE Healthcare (Pittsburgh, PA, USA); and the enzymes endo-polygalacturonase M2, endo-1,4-β-galactanase, and endo-1,5-α-L-arabinanase from Megazyme (Bray, Ireland). All the other chemicals were of analytical grade and commercially available or produced in China.

### 4.2. Polysaccharide Extraction and Purification

We extracted *A*. *Continentalis* using distilled water (material/dH_2_O, 1:20 [*w/v*]) twice at 100 °C for 3 h. We then concentrated the extracts under vacuum at 60 °C and precipitated them using 70% ethanol. After centrifugation (4000 rpm, 15 min), we collected the precipitate, redissolved it in water, and lyophilized it. Thus, we obtained three water-soluble polysaccharide fractions from the roots, stems, and leaves of *A. continentalis*, named WACP(R), WACP(S), and WACP(L), respectively. We then dissolved each water-soluble polysaccharide in dH_2_O and transferred the solution to a DEAE-cellulose column (6.0 × 20 cm, Cl^−^). We eluted the column with dH_2_O to obtain the neutral polysaccharides (WACP(R)-N, WACP(S)-N, and WACP(L)-N) and 0.5 M NaCl to obtain the acidic polysaccharides (WACP(R)-A, WACP(S)-A, and WACP(L)-A). Next, we purified the acidic polysaccharides in a Sepharose CL-6B column, which produced the major acidic polysaccharide fractions WACP(R)-A-c, WACP(S)-A-c, and WACP(L)-A-d.

### 4.3. Chemical Composition Analysis

We measured the total carbohydrate content by the phenol-sulfuric acid method using a mixture of major monosaccharides as a reference [36]. We measured the uronic acid content using the *m*-hydroxydiphenyl method with galacturonic acid (GalA) as a reference [37]. Finally, we measured the protein content using the Coomassie brilliant blue method [38].

### 4.4. Monosaccharide Composition and Molecular Weights Analysis

We first hydrolyzed the polysaccharide samples (2 mg) at 80 °C for 12 h using anhydrous methanol that contained 1 M HCl and then with 2 M trifluoroacetic acid (TFA) at 120 °C for 1 h. We then derived the released monosaccharides using HPLC following 1-phenyl-3-methyl-5-pyrazolone (PMP) pre-column derivatization [39]. We determined the molecular weights using high-performance gel permeation chromatography (HPGPC) with a TSKgel G-3000 PWXL column (7.8 × 300 mm) (Tosoh Corporation, Tokyo, Japan) coupled to a Shimadzu (Kyoto, Japan) HPLC system.

### 4.5. Fourier Transform Infrared Spectroscopy

We ground the polysaccharide samples with KBr powder and then compressed them into 1 mm pellets for Fourier transform infrared (FTIR) measurements. The detection range of the infrared spectrum was from 4000 cm^−1^ to 500 cm^−1^.

### 4.6. Enzymatic Hydrolysis Analysis

We dissolved the polysaccharides in 10 mM acetate buffer to 5 mg/mL. We then reacted three enzymes, including endo-polygalacturonase M2, endo-1,4-β-galactanase, and endo-1,5-α-L-arabinanase, with WACP(R)-A-c, WACP(S)-A-c, and WACP(L)-A-d under the following conditions: mass ratio of enzyme to substrate agent of 1:10, reaction temperature of 37 °C, continuous agitation, and a 24 h duration of the reaction. After the inactivation of the enzyme and centrifugation, we analyzed the enzymatic hydrolysates for changes in molecular weight, which we determined using gel-permeation chromatography on a TSK-gel G-3000PW XL column (7.8 × 300 mm) (Tosoh Corporation) coupled to a Shimadzu HPLC system.

### 4.7. NMR Spectroscopy

We recorded the ^13^C NMR spectra at 20 °C on a Bruker Avance 600 MHz spectrometer with a Bruker 5 mm broadband probe (Bruker, Billerica, MA, USA) operating at 150 MHz for ^13^C NMR. We dissolved the samples (20.0 mg) in D_2_O (0.5 mL) and centrifuged them to remove any undissolved polysaccharides. We analyzed the data using standard Bruker software.

### 4.8. Antioxidant Activity Analysis

#### 4.8.1. DPPH Radical Scavenging Activity

We measured the ability of pectic polysaccharides to scavenge DPPH radicals using a previously described method with some modifications [40]. We added polysaccharide sample solutions (0.6 mL) at different concentrations (0.5, 1, 2, 5, and 10 mg/mL) to 2.4 mL of 0.004% DPPH, which was dissolved in ethanol. We then incubated the mixture for 30 min without light and determined the absorbance at 517 nm. We used ascorbic acid as the positive control. *A*_0_ refers to the negative control, which was measured using dH_2_O instead of polysaccharides. *A*_1_ is the absorbance of the experimental group, and we measured *A*_2_ using ethanol instead of DPPH. We calculated the DPPH radical-scavenging activity as follows:$$\mathrm{DPPH}\;\mathrm{radical}\;\mathrm{scavenging}\;\mathrm{activity}\;\%\;=\;1-\frac{A_{1}-A_{2}}{A_{0}}\;\times 100\%$$

#### 4.8.2. Hydroxyl Radical Scavenging Activity

We measured the hydroxyl radical-scavenging activity of the polysaccharides according to a previously described method with some modifications [41]. We added polysaccharide sample solutions (0.5 mL) at different concentrations (0.5, 1, 2, 5, and 10 mg/mL) to 1.0 mL of 9 mM FeSO_4_, 1.0 mL of 9 mM salicylic acid (dissolved in ethanol), and 1 mL of hydrogen peroxide (H_2_O_2_). We then incubated the mixture at 25 °C for 30 min. We determined the absorbance at 510 nm. We used ascorbic acid as the positive control. *A*_0_ refers to the negative control, which was measured using dH_2_O instead of polysaccharides. *A*_1_ is the absorbance of the experimental group, and we measured *A*_2_ using dH_2_O instead of H_2_O_2_. We calculated the hydroxyl radical-scavenging activity as follows:$$\;\mathrm{Hydroxyl}\;\mathrm{radical}\;\mathrm{scavenging}\;\mathrm{activity}\;\%\;=\;1-\frac{A_{1}-A_{2}}{A_{0}}\;\times 100\%$$

#### 4.8.3. Superoxide Anion Radical Scavenging Activity

We measured the ability of the polysaccharides to scavenge superoxide anion radicals based on a previously described method with minor modifications [42]. We added polysaccharide sample solutions (50 µL) at different concentrations (0.5, 1, 2, 5, and 10 mg/mL) to 50 µL of 300 μM nitroblue tetrazolium (NBT), 50 µL of 936 μM reduced nicotinamide-adenine dinucleotide (NADH), and 50 µL of 120 μM phenazine methosulfate (PMS). We then incubated the mixture for 30 min without light. We determined the absorbance at 570 nm. We used ascorbic acid as the positive control. *A*_0_ refers to the negative control, which was measured using dH_2_O instead of polysaccharides. *A*_1_ is the absorbance of the experimental group, and we measured *A*_2_ using dH_2_O instead of PMS. We calculated the superoxide anion radical-scavenging activity as follows:$$\;\mathrm{Superoxide}\;\mathrm{anion}\;\mathrm{radical}\;\mathrm{scavenging}\;\mathrm{activity}\;\%\;=\;1-\frac{A_{1}-A_{2}}{A_{0}}\;\times 100\%$$

### 4.9. Statistical Analysis

We expressed the antioxidant activity as the mean ± standard deviation (SD) and performed all the tests on three sets of parallel experiments. We analyzed the data for significance using SPSS 22.0 (IBM, Inc., Armonk, NY, USA) with a *p* value of <0.05 considered statistically significant.

## 5. Conclusions

In summary, we compared the antioxidant activity, sugar composition, and molecular weight of the polysaccharides extracted from the roots, stems, and leaves of *A. continentalis*. The monosaccharide composition analysis indicated that the polysaccharides from the roots and stems were more similar to each other than they were to those from the leaves. The in vitro antioxidant experiment showed that the acidic polysaccharide fractions of *A. continentalis* had stronger antioxidant activity than the neutral fractions. Furthermore, we investigated the three purified acidic polysaccharide fractions (WACP(R)-A-c, WACP(S)-A-c, and WACP(L)-A-d) using NMR and enzymatic analysis. The structural analysis indicated that WACP(R)-A-c contained HG domains, WACP(S)-A-c contained HG and minor RG-II domains, and WACP(L)-A-d contained HG, RG-II, and RG-I domains. To our knowledge, this represents the first report on the structure and antioxidant activity of polysaccharides from different parts of *A. continentalis*. The results provide a theoretical basis for further applications of the roots, stems, and leaves of *A. continentalis* and will facilitate the development of *A. continentalis* polysaccharides for use in functional foods and medicine.

## Figures and Tables

**Figure 1 pharmaceuticals-15-01545-f001:**
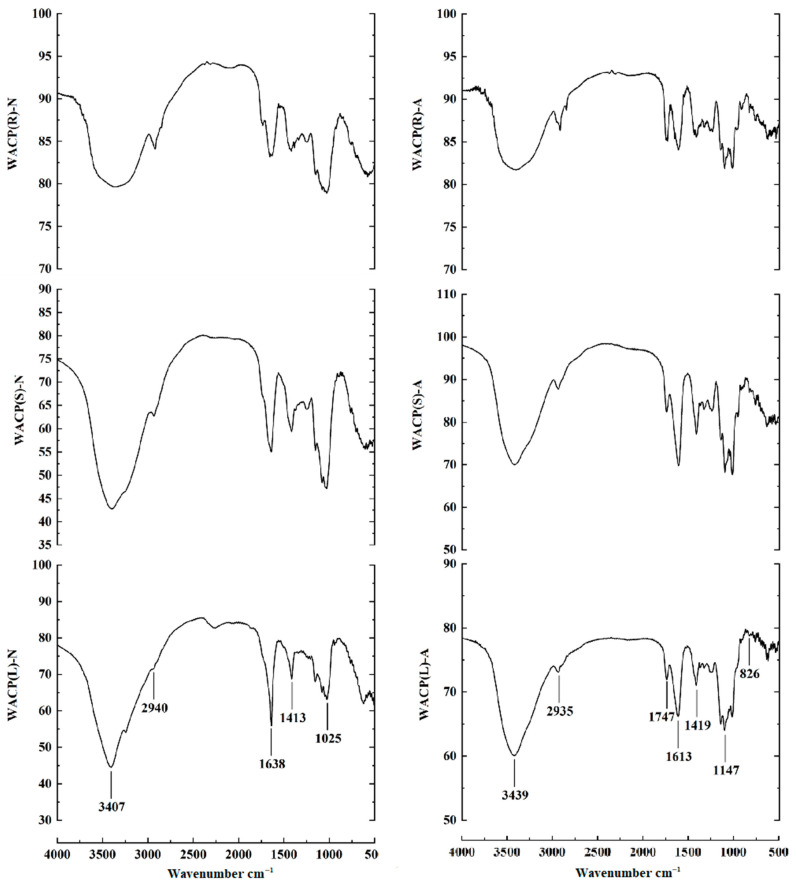
FTIR spectra of the polysaccharide fractions. (WACP(R)-N, WACP(S)-N, WACP(L)-N, WACP(R)-A, WACP(S)-A and WACP(L)-A).

**Figure 2 pharmaceuticals-15-01545-f002:**
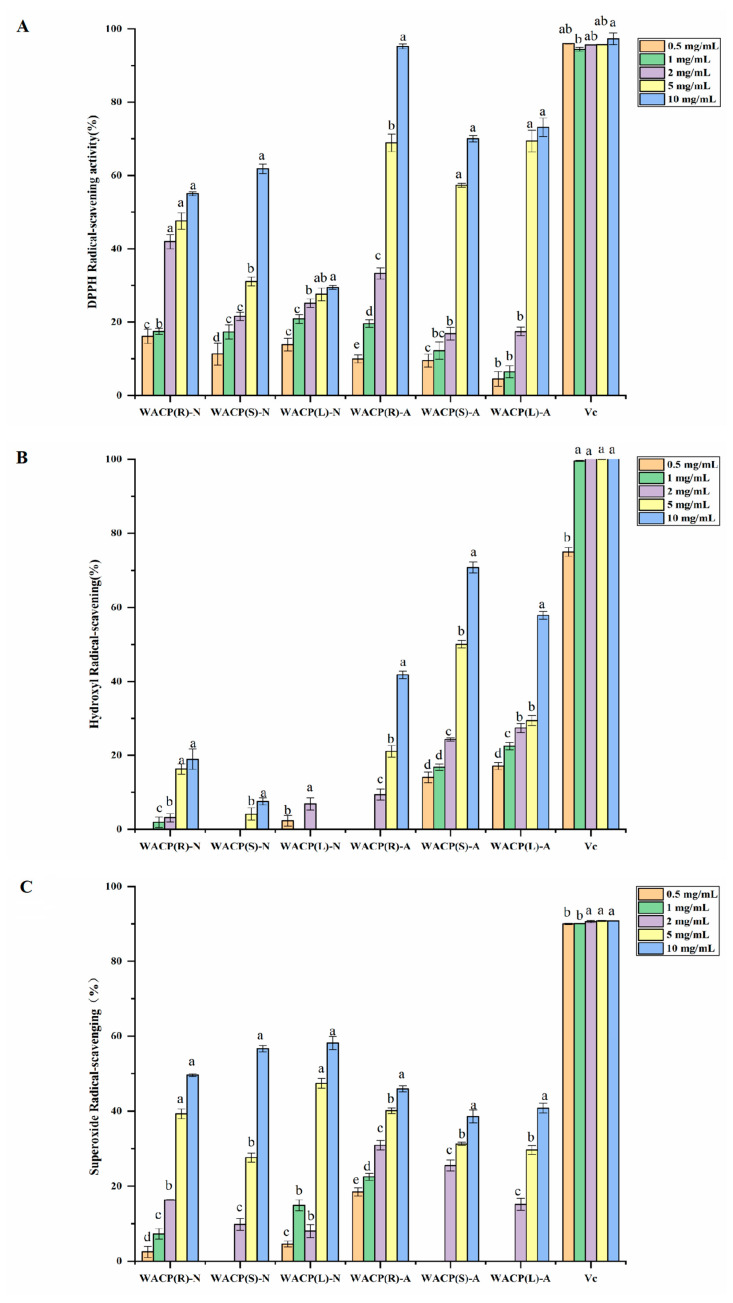
Scavenging ability of polysaccharide fractions on (**A**) DPPH radicals, (**B**) hydroxyl radicals and (**C**) superoxide anions. Vc was used as the positive control. Values are presented as means ± SD (n = 3). The same letter indicates that there was no significant change between different concentrations in the same group, and different letters indicate that there were significant differences between different concentrations in the same group.

**Figure 3 pharmaceuticals-15-01545-f003:**
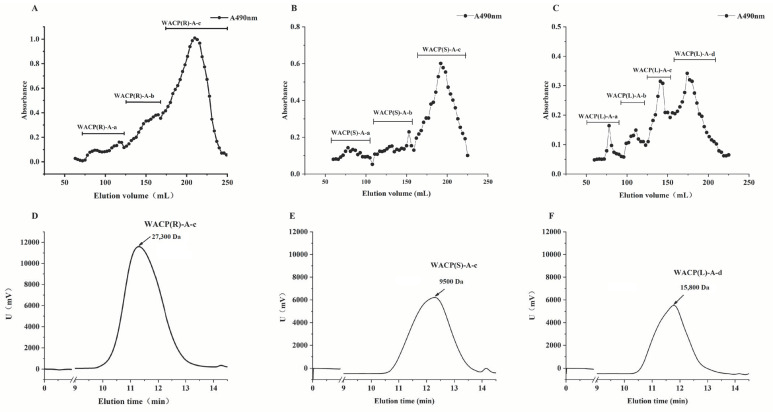
Purification of the pectic polysaccharide fractions ((**A**): WACP(R)-A, (**B**): WACP(S)-A, (**C**): WACP(L)-A) from *Aralia continentalis* and analysis by HPGPC ((**D**): WACP(R)-A-c, (**E**): WACP(S)-A-c, (**F**): WACP(L)-A-d).

**Figure 4 pharmaceuticals-15-01545-f004:**
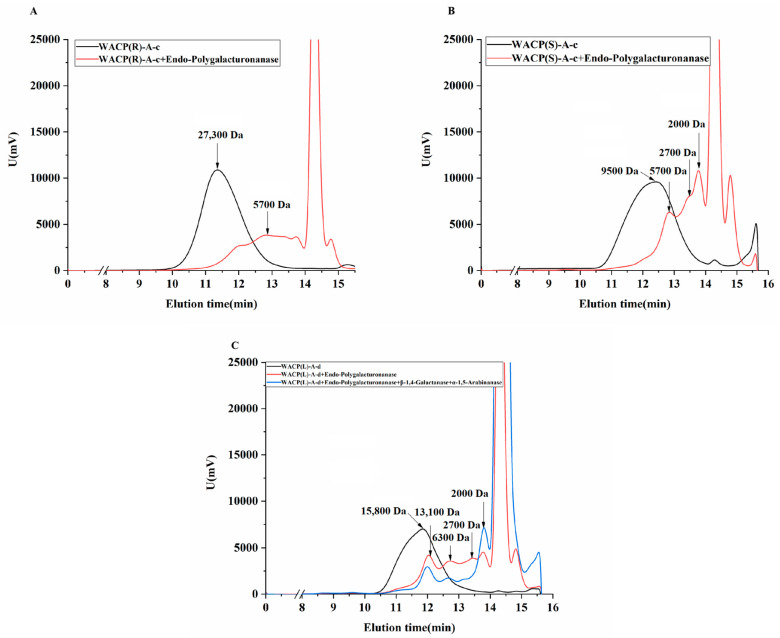
Enzymatic hydrolysis of the polysaccharides and HPGPC analyses.((**A**): WACP(R)-A-c, (**B**): WACP(S)-A-c, (**C**): WACP(L)-A-d).

**Figure 5 pharmaceuticals-15-01545-f005:**
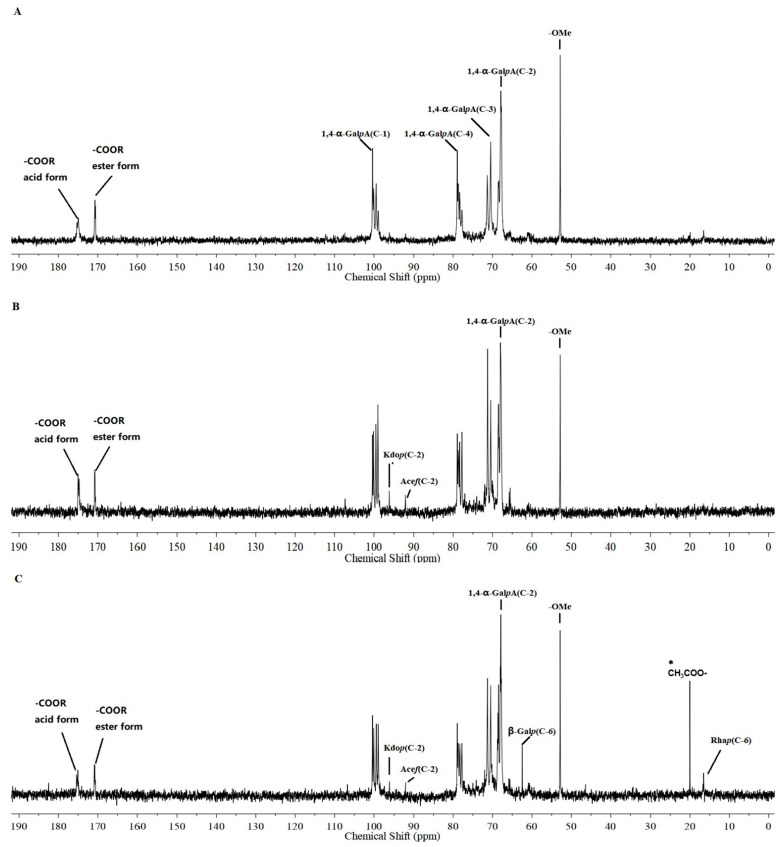
^13^C NMR spectra of (**A**) WACP(R)-A-c, (**B**) WACP(S)-A-c, and (**C**) WACP(L)-A-d. * means that the chemical shift of methyl carbon in CH3COO-.

**Table 1 pharmaceuticals-15-01545-t001:** Chemical characteristics and monosaccharide composition of the polysaccharides extracted from three different tissues of *Aralia continentalis*.

	WACP (R)	WACP (S)	WACP (L)
Yield (%)	3.6	12.9	8.2
Total sugar (%)	68.5	43.9	45.9
Protein (%)	1.5	3.2	1.1
Uronic acid (%)	30.2	50.0	24.5
Ash (%)	10.2	14.0	12.8
Sugar composition (%)			
Glucose (Glc)	34.5	28.5	44.7
Galactose (Gal)	21.4	16.9	11.7
Mannose (Man)	-	-	-
Rhamnose (Rha)	4.0	-	-
Glucuronic acid (GlcA)	-	-	-
Galacturonic acid (GalA)	23.5	25.71	14.7
Arabinose (Ara)	15.2	10.55	3.7
Fucose (Fuc)	-	-	-

Note: WACP, water-soluble polysaccharide-enriched fractions; L, leaves; R, roots; S, stems.

**Table 2 pharmaceuticals-15-01545-t002:** Monosaccharide composition of six subfractions isolated from three tissues of *Aralia continentalis*.

Fractions	Yield (%)	Monosaccharide Composition (Molar%)						
		Glc	Gal	Man	Rha	GlcA	GalA	Ara
WACP(R)-N	40.8	44.0	21.9	-	-	-	1.5	18.1
WACP(S)-N	41.2	38.5	32.6	2.3	-	-	4.0	16.3
WACP(L)-N	32.2	67.4	13.1	1.8	-	1.1	-	7.2
WACP(R)-A	59.2	-	10.0	-	-	-	71.2	10.1
WACP(S)-A	58.8	2.4	11.0	-	6.6	2.7	65.3	8.3
WACP(L)-A	67.8	3.2	26.7	5.0	7.8	5.4	36.1	15.7

Note: Ara, arabinose; Gal, galactose; GalA, galacturonic acid; Glc, glucose; GlcA, glucuronic acid; Man, mannose; Rha, rhamnose; WACP, water-soluble polysaccharide-enriched fractions.

**Table 3 pharmaceuticals-15-01545-t003:** Monosaccharide composition of three purified polysaccharide fractions from the acidic polysaccharides of *Aralia continentalis*.

Fractions	Yield (%)	Monosaccharide Composition (Molar%)						
		Glc	Gal	Man	Rha	GlcA	GalA	Ara
WACP(R)-A-c	75.3	1.0	5.3	-	3.0	1.0	84.5	4.5
WACP(S)-A-c	76.6	3.0	5.5	-	3.1	1.1	83.0	4.4
WACP(L)-A-d	46.6	10.1	19.3	-	11.5	2.1	44.8	12.3

## Data Availability

Data is contained within the article.

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
