# Peer review of "Analysis of Structure and Antioxidant Activity of Polysaccharides from Aralia continentalis"

_pharmaceuticals, 2022, doi:10.3390/ph15121545_

Round 1

Reviewer 1 Report

The article is novel and interesting. the language is very clear and understandable. However, minor changes are required before final publications  

1-     The following articles should be cited

https://bmccomplementmedtherapies.biomedcentral.com/articles/10.1186/s12906-018-2417-0

moreover, similar work for different plants should be cited

https://www.frontiersin.org/articles/10.3389/fnut.2022.976607/full

                 https://pubmed.ncbi.nlm.nih.gov/31805322/

additionally,

https://www.sciencedirect.com/science/article/pii/S2666351122000031

https://www.ncbi.nlm.nih.gov/pmc/articles/PMC6950075/

2-     Introduction line 36, names of biological activities details should be stated.

3-     Abbreviation list is strongly recommended

4-     Future research and study limitation should be suggested .

Best wishes

Author Response

Dear editor and reviewers:

Thank you very much for your helpful comments. Based on your comments, we have revised the manuscript. Followings are itemized answers to the comments.

  1. The following articles should be cited

https://bmccomplementmedtherapies.biomedcentral.com/articles/10.1186/s12906-018-2417-0 

moreover, similar work for different plants should be cited

https://www.frontiersin.org/articles/10.3389/fnut.2022.976607/full 

https://pubmed.ncbi.nlm.nih.gov/31805322/ additionally, https://www.sciencedirect.com/science/article/pii/S2666351122000031 https://www.ncbi.nlm.nih.gov/pmc/articles/PMC6950075/ 

Answer: We have referred those articles in the text. (references 8, 12, 14, 27, 28) 

  1. Introduction line 36, names of biological activities details should be stated.

Answer: Thank you for your suggestion, We have added the names of biological activities details in the part of introduction. (Line 36)

  1. Abbreviation list is strongly recommended

Answer: We have added abbreviation list in the text. (Lines 339-342)

  1. Future research and study limitation should be suggested.

Answer: We have added future research and study limitation in the discussion. (Lines 221-227)

Reviewer 2 Report

1. The title of the MS needs to be paraphrased since it contains confusing phrases, e.g. "active antioxidant" and "structural characterisation."

2. The MS contained a lack of novelty. No new biological targets or biomarkers were tested against the polysaccharides; in fact, it was only a conventional antioxidant assay. There were also no new compounds or derivatives reported from the A. continentalis.

3. the chemical characterisation of the polysaccharides were no convincing

4. The discussion was very brief and did not elaborate on the findings in the current literature

5. The conclusion did not emphasise the significant contribution of this research to the current literature.

Author Response

Dear editor and reviewers:

    Thank you very much for your helpful comments. Based on your comments, we have revised the manuscript. Followings are itemized answers to the comments.

Comments and Suggestions for Authors

  1. The title of the MS needs to be paraphrased since it contains confusing phrases, e.g. "active antioxidant" and "structural characterisation."

Answer: Thank you for your suggestion, we have revised the title.

  1. The MS contained a lack of novelty. No new biological targets or biomarkers were tested against the polysaccharides; in fact, it was only a conventional antioxidant assay. There were also no new compounds or derivatives reported from the continentalis.

Answer: In this study, authors compared the structure and biological activity of polysaccharides from different parts of A. continentalis, the research determined that the stem of A. Continentalis could be a potential functional food and medicine instead of the root of A. Continentalis, which was mainly a traditional Chinese medicine. So only the basic structural analysis and antioxidant activity in vitro were explored.

  1. the chemical characterisation of the polysaccharides were no convincing

Answer: The chemical characterisation of the polysaccharides were analyzed by the methods of NMR and Enzymolysis analysis, which could identify structural of  differences polysacchrides from A. continentalis.

  1. The discussion was very brief and did not elaborate on the findings in the current literature.

Answer: Thank you for your suggestion, the analysis and comparison with current literatures are added in the discussion section.

  1. The conclusion did not emphasise the significant contribution of this research to the current literature.

Answer: we have added the significant contribution of this research in the conclusion. (Lines 334-338)

Reviewer 3 Report

 Authors present an interesting study about polysaccharides extracted from different parts (root, stem, and leaf ) of  Aralia Continentalis  tree.  The results, their discussion, and methodological design are presented in the right way. Polysaccharides extraction, purification, and structural analysis presented from the different types of polysaccharides are very well documented. I only recommend improving the methodological information about liquid chromatographic conditions, used in monosaccharides composition, for example, mobile phase composition, discussion about the retention time of sugars, and so on could be good data for readers.

In line 11, the word three must be changed to tree

Author Response

Dear editor and reviewers:

Thank you very much for your helpful comments. Based on your comments, we have revised the manuscript. Followings are itemized answers to the comments.

Comments and Suggestions for Authors

Authors present an interesting study about polysaccharides extracted from different parts (root, stem, and leaf ) of  Aralia Continentalis tree.  The results, their discussion, and methodological design are presented in the right way. Polysaccharides extraction, purification, and structural analysis presented from the different types of polysaccharides are very well documented. I only recommend improving the methodological information about liquid chromatographic conditions, used in monosaccharides composition, for example, mobile phase composition, discussion about the retention time of sugars, and so on could be good data for readers.

Answer: Thank you for your suggestion, we will improving the methodological information about liquid chromatographic conditions in our further research.

  1. In line 11, the word three must be changed to tree

Answer: To better understand this, we have revised this sentense in the text. (Line 12)

Round 2

Reviewer 2 Report

ok

Author Response

Dear reviewer:

Thank you very much for your helpful comments. We have finished the extensive English revisions, and revised the manuscript according to your comments. Followings are itemized answers to the comments.

Reviewer 2:

Comments and Suggestions for Authors

Ok

Answer: Thanks again for your suggestions